# No Changes in Body Composition and Adherence to the Mediterranean Diet after a 12-Week Aerobic Training Intervention in Women with Systemic Lupus Erythematosus: The EJERCITA-LES Study

**DOI:** 10.3390/nu15204424

**Published:** 2023-10-18

**Authors:** Blanca Gavilán-Carrera, Alba Ruiz-Cobo, Francisco José Amaro-Gahete, Alberto Soriano-Maldonado, José Antonio Vargas-Hitos

**Affiliations:** 1Departamento de Medicina Interna, Hospital Universitario Virgen de las Nieves, 18014 Granada, Spain; joseantoniovh@hotmail.com; 2Instituto de Investigación Biosanitaria ibs.Granada, 18012 Granada, Spain; amarof@ugr.es; 3PA-HELP “Physical Activity for HEaLth Promotion” Research Group, University of Granada, 18071 Granada, Spain; 4Sport and Health University Research Institute (iMUDS), University of Granada, 18071 Granada, Spain; 5Department of Physiology, Faculty of Medicine, University of Granada, 18001 Granada, Spain; albaruizcobo@correo.ugr.es; 6CIBER de Fisiopatología de la Obesidad y Nutrición (CIBEROBN), Instituto de Salud Carlos III, 28029 Granada, Spain; 7Department of Education, Faculty of Education Sciences, University of Almería, 04120 Almería, Spain; asoriano@ual.es; 8SPORT Research Group (CTS-1024), CERNEP Research Center, University of Almería, 04120 Almería, Spain

**Keywords:** obesity, diet, autoimmune diseases, exercise

## Abstract

Systemic Lupus Erythematosus (SLE) is a chronic autoimmune disease linked to high cardiovascular risk. To reach an adequate body composition status while maintaining proper dietary habits are effective strategies for reducing cardiovascular risk, both being potentially modified through exercise. This study aimed to evaluate the effects of a 12-week aerobic training intervention on anthropometry, body composition and adherence to the Mediterranean diet in women with SLE. A total of 58 women with SLE were assigned to either an exercise group (EG; *n* = 26) or a comparison group (CG; *n* = 32) in this non-randomized controlled trial. The EG comprised 12 weeks of aerobic exercise (two sessions/week) between 40–75% of the individual’s heart rate reserve (calculated as maximum heart rate − resting heart rate) and the CG received usual care. At baseline and after the intervention, the anthropometry (i.e., weight, waist circumference, waist-to-hip ratio, and body mass index) and body composition (i.e., fat mass and lean mass) were assessed using a stadiometer, an anthropometric tape, and a bioimpedance device, respectively. Dietary habits were assessed with the Mediterranean Diet score. There were no between-group differences in neither anthropometric nor body composition parameters (all *p* > 0.05). Similarly, no between-group differences were obtained in the adherence to the Mediterranean diet after the exercise intervention (all *p* > 0.05). Contrary to the initial hypothesis, these results suggest that the 12-week aerobic training intervention performed in this study did not improve anthropometry, body composition or adherence to the Mediterranean diet in women with SLE.

## 1. Introduction

Systemic lupus erythematosus (SLE) is a chronic inflammatory and autoimmune disease with heterogeneous clinical manifestations depending on the organs and systems affected [1]. The global SLE prevalence is estimated to be 44 per 100,000 inhabitants, with young adult women being the most affected by this disease [2]. Although the disease’s overall prognosis has improved in recent years, patients with SLE present an increased cardiovascular (CV) risk, and CV diseases represent the main cause of morbidity and mortality in this population [3]. Consequently, it is clinically relevant to identify CV risk factors in the SLE and potential alternatives for their prevention and management.

Obesity—a metabolic disease which is present in 28–50% of SLE patients [4]—leads to the development of CV disease and mortality independently of other CV risk factors [5]. Beyond the conventional obesity indicators (i.e., body mass index [BMI] and fat percentage), markers of central adiposity (i.e., waist circumference or waist-to-hip ratio) have shown a particular harmful association with CV diseases [5]. In addition, the combination of obesity and reduced muscle mass can synergistically contribute to the development of metabolic disorders related to lipid profile and overall CV risk [6]. In this regard, exercise has emerged as a powerful strategy to optimize body composition [7,8,9] becoming one of the most promising non-pharmacological therapies to improve health in SLE [10].

To date, several studies have examined the effect of different exercise interventions on anthropometric and body composition parameters in SLE patients [11,12,13,14], with no significant effects on these dimensions according to a recent meta-analysis [15]. However, the low number of studies and their small sample size limit the establishment of robust conclusions on this matter. Furthermore, the exercise interventions carried out in the above-mentioned studies [11,12,13,14] are also insufficiently detailed for replication and fail to provide accurate information on monitoring training load, both points being essential for the optimal translation of evidence into clinical practice [16]. This may be particularly relevant in the field of SLE, where disease-specific exercise recommendations are still lacking. In this context, most conservative recommendations of physical activity for the general population (i.e., 150 min of moderate physical activity per week according to the American College of Sports Medicine (ACSM) [17]) could be reasonable advice delivered in clinical settings. The potential of this exercise dose to modify relevant markers of body composition in patients with SLE is yet to be determined.

Dietary habits are key modulators of CV risk in SLE [18]. More specifically, the Mediterranean diet has gained growing attention due to its association with CV protection [19]. The Mediterranean diet is characterized by whole or minimally processed foods and a high intake of vegetables, fruits, whole grains, fish and olive oil, with moderate consumption of red meat and wine [20]. A greater adherence to the Mediterranean diet has been associated with lower disease activity and CV risk in SLE patients [21], yet there have been no intervention studies aimed at enhancing it. Interestingly, promoting change in one health behavior can indirectly lead to the adoption of other health behaviors, known as the “transfer effect” [22]. In this sense, physical activity interventions have also evidenced changes in dietary preferences (such as increased fruit and vegetable intake [23,24,25,26]) enclosed in the Mediterranean diet. However, it is unclear to which extent an exercise intervention could specifically influence adherence to this dietary pattern in patients with SLE.

Therefore, the aim of the present study was to evaluate the effects of a 12-week aerobic training intervention following the ACSM guidelines on anthropometry, body composition and adherence to the Mediterranean diet in women with SLE. It was hypothesized that the intervention would improve anthropometry, body composition [27] and adherence to the Mediterranean diet [23,24,25,26] in women with SLE.

## 2. Material and Methods

On 11 April 2017, the protocol for this non-randomized controlled trial was registered on clinicaltrials.gov [NCT03107442]. This process was carried out before the enrolment of participants started.

### 2.1. Participants

The participants were selected through a telephone screening process at the Systemic Autoimmune Diseases Unit of the “Virgen de las Nieves” and “San Cecilio” University Hospitals in Granada. The inclusion criteria were women: (i) diagnosed with SLE according to the criteria of the American College of Rheumatology [28], (ii) with at least 1 year of follow-up in the units, (iii) stability in treatment and medical history, and (iv) not regularly engaged in exercise. Exclusion criteria were women: (i) on biologic treatment in the last six months or requiring a dose of prednisone >10 mg/day, (ii) with recent personal history of CV disease in the past year, (iii) with contraindications for exercising, (iv) suffering from other associated rheumatic conditions, active acute or chronic infections, neoplasms, renal failure, any cardiac or pulmonary condition, (v) having a BMI > 35 kg/m^2^ (to guarantee proper measurement of arterial stiffness, main outcome of the project [29]), or (vi) experiencing cognitive impairment. Before the study, all patients received information about the procedures and signed a written consent form. The Granada Research Ethics Committee approved the protocol on 11 November 2016 (reference No.:10/2016).

### 2.2. Procedure

An initial screening of participants was conducted via telephone. Potentially eligible patients were invited for an in-person visit, and if they met the criteria, the first assessment day took place. Data collection occurred over 2 days. On the first day, anthropometry and body composition were assessed, sociodemographic and clinical information was collected, and patients were given questionnaires to complete at home. Two to four days later, the second assessment day was conducted, during which the research team reviewed the questionnaires to minimize missing data.

### 2.3. Interventions

#### 2.3.1. Comparison Group

After the initial assessment, the patients assigned to the Comparison Group (usual care) were given basic guidelines to adopt a healthy lifestyle as normally delivered in standard treatment. This included basic nutritional recommendations (i.e., reducing processed foods and increasing intake of vegetables/fruits) and physical activity advice (i.e., reducing sedentary behavior and increasing daily levels of physical activity).

#### 2.3.2. Experimental Group

Patients assigned to the exercise group received identical guidelines for adopting a healthy lifestyle (i.e., basic nutritional recommendations and advice on physical activity), just like those in the comparison group. In addition, they participated in two 75-min sessions per week during the whole intervention (i.e., 12 weeks) involving aerobic exercise on a treadmill (BH, Series, i.RC12 Dual, Vitoria-Gasteiz, Spain) at moderate to vigorous intensity. The exercise program’s details are available elsewhere [29]. Importantly, the Consensus of Exercise Reporting Template was strictly followed [16]. In summary, the weekly volume of moderate-to-vigorous physical activity (MVPA) gradually increased from 90–145 min in weeks 1–4 while 150 min were reached in weeks 5–12. The program transitioned from continuous exercise (weeks 1–6), to continuous plus interval exercise (weeks 6–8), finishing with only interval exercises (weeks 9–12).

Sessions took place in groups of five patients in a quiet room at the “Virgen de las Nieves” Hospital, Granada (Spain). Accredited sport sciences staff and Internal Medicine doctors supervised all sessions. Attendance was recorded daily, and patients were contacted if they missed a session to determine the reason and encourage makeup. Adherence to exercise is reported as (i) the median attendance frequency and (ii) the proportion of patients attending more than 75% (i.e., 18 sessions, the minimum required for assessment of efficacy) and (iii) more than 90% of the sessions.

Each session included a 3–4-min warm-up on the treadmill at 35–40% of the heart rate reserve (HRR) and 3–4 min of active stretching of major muscle groups and ended with a cooling down phase of major muscle groups’ static stretching. Exercise was individually prescribed, with training intensity ranging from 40 to 75% of each individual’s HRR. The maximum heart rate (HRmax) was estimated as 208 − (0.7 × age) [30]. The training (or target) heart rate (tHR) was calculated with the formula tHR = resting HR + (%HRR). Heart rate was continuously monitored during all sessions (Polar V800, Kempele, Finland), and the session rating of perceived exertion was additionally used [31].

Potential adverse effects, such as CV symptoms or signs (e.g., chest pain, dizziness, arrhythmia, or effort-related hypotension), musculoskeletal injuries, and falls were successfully minimized through (i) a treadmill familiarization phase, (ii) proper warm-up and cool down phases in all sessions, and (iii) active supervision by the research team which included periodic HR and symptoms monitoring.

### 2.4. Dependent Outcomes

In a prior study from this non-randomized controlled trial [29], the primary outcome (arterial stiffness) has been published. The following are secondary analyses of the exercise intervention, encompassing various measures of anthropometry, body composition and adherence to the Mediterranean diet.

#### 2.4.1. Anthropometry and Body Composition

Weight, fat mass, and lean mass (kg) were estimated using a bioimpedance device (InBody R20, Biospace, Seúl, Corea). Height (cm) was measured with a height gauge (SECA 222, Hamburgo, Alemania), and BMI subsequently calculated (weight in kg/height in m^2^). Waist and hip circumferences (cm) were determined with an anthropometric tape (Harpenden, Holtain Ltd., Wales, UK), and waist-to-hip ratio (waist circumference/hip circumference) calculated.

#### 2.4.2. Adherence to the Mediterranean Diet

Adherence to the Mediterranean diet was estimated using the Mediterranean Diet Score [32]. It consists of 11 items including whole grains, fruits, vegetables, potatoes, legumes, fish, poultry, dairy products (such as cheese, yogurt, and milk), red meats and their derivatives, olive oil, and red wine. Based on the portions the patients had consumed, the foods were scored from 0 to 5 (according to their position in the Mediterranean Diet pyramid). Total score ranges from 0 to 55, where higher scores indicate higher adherence to the Mediterranean diet and greater diet quality.

#### 2.4.3. Other Assessments

Participants completed a questionnaire on sociodemographic and clinical data. Disease progression was assessed using the Systemic Lupus Erythematosus Disease Activity Index (SLEDAI), with a range of 0–105 where a high score indicates a higher degree of disease activity [33].

### 2.5. Patient Allocation and Blinding

Different factors, primarily the distance from the hospital, as well as other personal reasons related to time availability (work and/or domestic responsibilities), prevented a significant number of participants from regularly attending the intervention over the 12-week period. Therefore, in order to maximize sample size, randomization was not finally possible. To minimize selection bias, the groups were matched by age (±2 years), BMI (±1 kg/m^2^), and SLEDAI (±1 unit). The researcher responsible for the selection process was blinded.

### 2.6. Statistical Analysis

The sample size was calculated for the primary outcome (i.e., arterial stiffness), as indicated in a previous study [29]. Normality was assumed due to sample size. Descriptive statistics (mean and standard deviation for quantitative variables, and frequencies and percentages for qualitative variables) were used to examine the sociodemographic and clinical characteristics of the sample. To compare baseline characteristics between groups, the student’s *t*-test for independent samples (quantitative variables) and the Chi-square test (qualitative variables) were used. Between-groups differences in changes from baseline at week 12 were analyzed using an analysis of covariance (ANCOVA), including the dependent variable at baseline as a covariate. To gain further insight into the compliance with the planned training intensity, paired T-tests were used to analyze potential differences between the minimum exercise HR planned for each session and interval, and the average HR recorded measured by HR monitors.

The primary analyses of this study were defined as “per protocol” to assess efficacy, and patients in the exercise group were included if attendance was ≥75%. The achieved power in these analyses was calculated for each of the outcomes (i.e., anthropometry, body composition and adherence to the Mediterranean diet) using the G*Power 3.1 program. Subsequently, two different sensitivity analyses were conducted to assess the robustness of the results: (i) imputation of missing values with the first observed data (intention-to-treat); (ii) per protocol with a minimum attendance of 90%.

All analyses were performed using the Statistical Package for Social Sciences (SPSS) version 23.0 (IBM Corp., Armonk, NY, USA). The significance level was set at *p* < 0.05.

## 3. Results

The flowchart of participants throughout the study can be seen in Figure 1. The median attendance frequency was 22.5 sessions (~94%). A total of 22 participants (~85%) attended at least 75% of the sessions (included in the primary analyses), and 18 participants (~69%) attended at least 90% of the sessions. One participant discontinued the intervention due to lack of time. A total of four participants were lost to follow-up in the comparison group, and none in the intervention group. No adverse effects were recorded during the intervention.

Table 1 displays the baseline characteristics of the patients at the beginning of the study. No significant differences were found between the exercise group and the comparison group for any outcome of interest before the intervention (for all, *p* > 0.05).

Appendix A shows the differences between the minimum planned HR for each participant in different sessions and intervals, and the average HR recorded during the sessions. During the first seven sessions, the average HR achieved was higher than the minimum planned HR, with observed mean differences (standard error) ranging from 3.1 (3.6) to 5.5 (1.8) beats per minute. For sessions 11, 13, 15, and 17–23, the average HR achieved was below the minimum planned HR, with observed mean differences ranging from 3.9 (5.9) to 9.3 (11.5) beats per minute.

The between-group differences for changes at the end of the intervention are presented in Table 2. There were no between-group differences in neither anthropometric nor body composition parameters (all, *p* > 0.05). Similarly, no between-group differences were obtained in the adherence to the Mediterranean diet after the exercise intervention (all, *p* > 0.05). The power achieved in the analyses ranged from 8% to 50%.

The sensitivity analyses, including the imputation of missing values (intention to treat), are shown in Table 3. The comparison group exhibited a greater reduction in waist circumference compared to the exercise group (mean difference = 4.1, 95% CI 0.5 to 7.7, *p* = 0.026). There were no between-group differences in other anthropometric or body composition parameters (all, *p* > 0.05). No between-group differences were obtained in the adherence to the Mediterranean diet after the exercise intervention (all, *p* > 0.05).

The sensitivity analyses, including a 90% attendance to the exercise intervention, are presented in Table 4. There were no between-group differences in neither anthropometric nor body composition parameters (all, *p* > 0.05). Similarly, no between-group differences were obtained in the adherence to the Mediterranean diet after the exercise intervention (all, *p* > 0.05).

## 4. Discussion

The main findings of the present study suggest that, contrary to the initial hypothesis, a 12-week aerobic training intervention following the ACSM recommendations did not improve anthropometry, body composition, or adherence to the Mediterranean diet in women with SLE.

To date, only four previous studies in patients with SLE have evaluated the effect of exercise on anthropometric and body composition parameters [11,12,13,14]. Benatti et al. [11] observed that a 12-week concurrent exercise intervention (i.e., strength and aerobic training [60 min per week of moderate intensity]) improved the BMI of patients with SLE. However, other studies that implemented a walking exercise program for 12 weeks (i.e., 60 to 100 min per week at moderate intensity) did not find any significant effect on weight, fat mass, lean mass, or abdominal fat [12,13]. In this sense, a previous study conducted an exercise program consisting of walking three days a week (120–180 min at light to moderate intensity) for 16 weeks obtaining no significant changes on BMI, waist circumference, or waist-to-hip ratio [14]. Despite the different training protocols used, the results of the present study concur with those reported by a recent meta-analysis [15] which showed a lack of effect on various anthropometric and body composition parameters in SLE patients. Although it was also found that the exercise group increased waist circumference compared to the comparison group, this result was not consistent across sensitivity analyses and could be possibly driven by the conservative data imputation approach taken. Furthermore, it is difficult to distinguish clinically relevant changes in waist circumferences from measurement errors due to large measurement errors related to this variable [34]. Despite the efforts made to standardize data collection in this study, extensive training is required to reduce measurement errors [34]. Therefore, the possibility of errors and discrepancies among assessors cannot be completely ruled out. Given that hip circumference measures might lead to less errors [35], this could explain the inconsistency in findings related to waist circumference vs. waist-to-hip ratio.

These findings contrast with compelling evidence demonstrating the beneficial effect of exercise intervention on improving anthropometric and body composition parameters in different populations [7,8,9]. Various reasons could explain these discrepancies. Firstly, the exercise interventions previously implemented in patients with SLE (including the one selected for this study) did not primarily aim to induce improvements in body composition [15]. For that purpose, a combination of aerobic training (moderate to vigorous intensity, 30 to 60 min or more daily) and strength training is highly recommended [17]. Indeed, we only found one study which reported a significant improvement of anthropometric parameters in SLE and this included a strength training component [11]. In this line, a recent meta-analysis has demonstrated the superiority of concurrent training over the rest of training modalities at improving body composition and inflammatory parameters in overweight and obese individuals [7]. Therefore, it seems plausible that the type (predominantly aerobic), volume (<180 min per week), and intensity (predominantly moderate) of training proposed in various exercise interventions for patients with SLE (including the one conducted here) may have been insufficient to induce improvements in body composition.

Moreover, due to the insufficient details provided in previous exercise interventions in SLE [11,12,13,14], it is unclear whether the participants adhered to the prescribed program in terms of attendance or intensity. In this regard, our results indicate that the intensity achieved by the participants was often lower than expected, a point that could explain the absence of effect on anthropometric and body composition parameters. Furthermore, exercise training could increase appetite and, consequently, energy intake [36]. Indeed, while the transfer hypothesis proposes that exercise leads to generalization of healthy behaviors [22], the compensation hypothesis assumes that exercise leads to increased calorie intake [22] and a decrease in other physical activities to “compensate” for the energy expenditure [37]. Because changes in anthropometry/body composition depend on the balance of energy intake vs. expenditure and data in this regard were not recorded, the possibility that the compensatory responses explained the lack of improvement of these outcomes remains open.

Pocovi-Gerardino et al. recently reported that greater adherence to the Mediterranean Diet was associated with improved disease-related parameters in patients with SLE [21]. To the best of our knowledge, no intervention studies have focused on enhancing adherence to the Mediterranean Diet in this population. Although exercise can indirectly modify dietary habits [22,23,24,25,26], the present study showed no influence of the exercise intervention on the adherence to the Mediterranean diet. In line with these findings, a previous work conducted in perimenopausal women [38] reported no changes in overall adherence to the Mediterranean Diet following a four-month concurrent exercise intervention. However, the study did reveal changes in certain food groups, such as increased fish and beer consumption after exercise [38]. It is worth mentioning that in the present study adherence to the Mediterranean diet was slightly and consistently enhanced in the exercise group, although this increase did not reach statistical significance. Given that previous research has shown that physical activity interventions can indirectly increase fruit and vegetable consumption [23,24,25,26], it is possible that exercise primarily influences specific food groups within the Mediterranean Diet, rather than the diet as a whole. Therefore, it could be hypothesized that: (i) our exercise intervention could be a stimulus that only produces slight but insufficient modifications in dietary patterns and/or (ii) Mediterranean Diet could be a very specific pattern to be indirectly modified through exercise. Further studies are needed to examine what types of lifestyle-based interventions could increase the adherence to the Mediterranean Diet and, simultaneously, improve body composition in patients with SLE.

### 4.1. Limitations and Strengths

Given the low statistical power achieved, the sample size may be insufficient to detect changes in the dependent outcomes of this work. Furthermore, this study only included women with mild or inactive SLE disease activity and BMI below 35 kg/m^2^; therefore, the results are not generalizable to other patient profiles (such as men, patients with moderate/high disease activity or obesity class 2–3). It is also important to note the lack of control over energy intake and expenditure, as changes in anthropometry and body composition are strongly influenced by this energy balance. Finally, while the absence of randomization could be seen as a limitation, the quality of the trial may exert a more significant influence on the treatment effect size than randomization alone [39]. High-quality non-randomized controlled studies can yield outcomes that closely resemble those observed in randomized controlled trials [39]. On the contrary, the study has important strengths that should be highlighted. Firstly, adherence to the exercise program was considerably high and accurately recorded and reported. Additionally, this intervention is described following the CERT guidelines [16] with the aim of enhancing transparency and replicability of exercise interventions, allowing any professional to implement it.

### 4.2. Future Research Directions

Future intervention studies may benefit from creating higher energy expenditure along with incorporating a strength training component to optimize improvements in body composition [7,9]. Also, controlling and reporting daily dietary intake along with physical activity outside the program will be needed to better explain changes in body composition due to energy deficit or surplus. In addition, the inclusion of a diet focused on generating an energy deficit or the incorporation of nutritional education sessions (which have proven effective for improving body composition in other populations [40]) should be explored as potential tools in patients with SLE. Reaching changes in body composition is challenging, but seem to be better addressed by frequent and sustained interventions [41], particularly with extended-duration interventions [42]. Then, analyzing the effect of longer intervention periods is warranted. Finally, the use of dual-energy X-ray absorptiometry or even magnetic resonance imaging (the gold standard method) in future research would allow for a more precise estimation of the anthropometric and body composition parameters analyzed in this study [43].

## 5. Conclusions

The main findings of the study suggest that a 12-week aerobic training intervention following ACSM recommendations did not improve anthropometry, body composition or adherence to the Mediterranean diet in women with SLE. Future multidisciplinary interventions are needed to understand which training parameters (type, intensity, duration, and frequency) and dietary interventions lead to successful changes in body composition and adherence to the Mediterranean diet in women with SLE.

## Figures and Tables

**Figure 1 nutrients-15-04424-f001:**
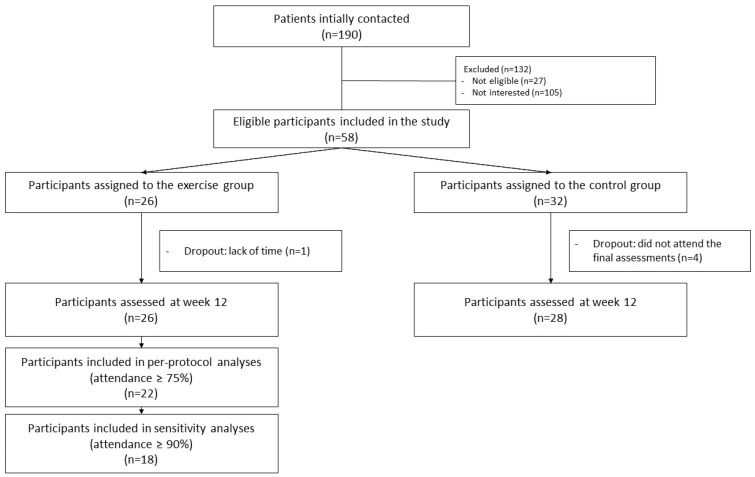
Flow chart of the study participants throughout the study.

**Table 1 nutrients-15-04424-t001:** Baseline descriptive characteristics of the study participants.

	All (*n* = 58)	Exercise (*n* = 26)	Comparison (*n* = 32)	*p*
	Mean (SD)	Mean (SD)	Mean (SD)	
Age, years	44.0 (13.9)	43.0 (15.1)	44.8 (13.1)	0.618
Disease duration, years	15.4 (10.5)	14.5 (10.4)	16.1 (10.6)	0.570
Marital status (single/married/divorced; %)	44.8/50.0/5.2	53.8/42.3/3.9	37.5/56.3/6.2	0.455
Educational level (No studies/primary/secondary/university; %)	3.4/36.3/22.4/37.9	0/38.5/26.9/34.6	6.3/34.4/18.7/40.6	0.521
Occupational status (working/housewife/not working; %)	41.4/24.1/34.5	42.3/19.2/38.5	40.6/28.1/31.3	0.706
Anthropometry				
Weight, kg	64.8 (11.9)	66.2 (8.7)	63.7 (14.0)	0.426
Waist circumference, cm	79.4 (10.9)	80.2 (9.7)	78.8 (11.8)	0.623
Body Composition				
Waist-to-hip ratio	0.83 (0.1)	0.84 (0.07)	0.83 (0.07)	0.391
Fat percentage (%)	33.9 (7.5)	35.9 (5.7)	32.2 (8.4)	0.059
Lean mass percentage (%)	35.8 (4.0)	34.8 (3.2)	36.6 (4.4)	0.094
BMI, kg/m^2^	25.2 (4.7)	25.9 (3.4)	24.7 (5.6)	0.336
Lean mass, kg	22.9 (2.8)	22.9 (2.7)	22.9 (3.0)	0.968
Adherence to the Mediterranean Diet (0–55)	29.4 (6.3)	29.2 (6.4)	29.5 (6.2)	0.887
SLEDAI (0–105)	0.22 (0.9)	0.04 (0.2)	0.38 (1.18)	0.158
SDI (0–46)	0.47 (1.11)	0.19 (0.63)	0.69 (1.35)	0.092
Immunosupressant intake (Yes/No; %)	44.8/55.2	46.2/53.8	43.8/56.2	0.855
Corticosteroid intake (Yes/No; %)	62.1/37.9	57.7/42.3	65.6/34.4	0.536
Hydroxicloroquine intake (Yes/No; %)	89.7/10.3	96.2/3.8	84.4/15.6	0.143
Smoke (Non-smoker/Current smoker/Former smoker; %)	63.8/25.9/10.3	76.9/15.4/7.7	53.1/34.4/12.5	0.166
Alcohol (Yes/No; %)	5.2/94.8	7.7/92.3	3.1/96.9	0.435
Diabetes (Yes/No; %)	1.7/98.3	0/100.0	3.1/96.9	0.363
Systolic Blood Pressure, mm/Hg	117.7 (10.3)	116.8 (10.0)	118.4 (10.6)	0.567
Diastolic Blood Pressure, mm/Hg	75.5 (9.5)	75.6 (8.8)	75.4 (10.1)	0.936
Mean Blood Pressure, mm/Hg	94.8 (8.8)	94.5 (8.3)	95.0 (9.3)	0.821
Dyslipidemia (Yes/No; %)	17.2/82.8	19.2/80.8	15.6/84.4	0.718
Menopause (Yes/No; %)	39.7/60.3	38.5/61.5	40.6/59.4	0.867

BMI: Body Masss Index; SLEDAI, systemic lupus erythematosus disease activity index; SDI, systemic damage index.

**Table 2 nutrients-15-04424-t002:** Per-protocol (primary) analyses assessing the effects of a 12-week aerobic training intervention on anthropometry, body composition and adherence to the Mediterranean diet in women with systemic lupus erythematosus (participants in the exercise group were included if attendance was ≥75%).

	Change from Baseline at Week 12 (Final–Baseline)							
	Exercise (*n* = 22)		Comparison (*n* = 28)		Exercise vs. Comparison			
	Mean	SE	Mean	SE	Mean Difference	(95% CI)		*p*
Weight (kg)	−0.9	0.5	−1.5	0.4	0.7	−0.6	1.9	0.297
BMI (kg/m^2^)	−0.4	0.2	−0.5	0.2	0.1	−0.4	0.6	0.604
Waist circumference (cm)	0.4	1.5	−3.5	1.3	3.9	−0.1	7.9	0.057
Waist-to-hip ratio	−0.04	0.01	−0.03	0.01	−0.01	−0.1	0.0	0.675
Fat percentage (%)	−1.8	0.8	−1.7	0.8	−0.1	−2.4	2.2	0.958
Lean mass (kg)	−0.1	0.4	0.2	0.3	−0.3	−1.3	0.7	0.569
Lean mass percentage (%)	0.4	0.6	1.1	0.5	−0.7	−2.4	0.9	0.375
Adherence to the Mediterranean Diet (0–55)	1.8	1.2	−0.5	1.1	2.2	−1.1	5.5	0.178

The analyses were adjusted for baseline values. BMI, Body Mass Index; CI, confidence interval; SE, standard error.

**Table 3 nutrients-15-04424-t003:** Sensitivity analyses: Baseline-observation carried forward imputation assessing the effects of a 12-week aerobic training intervention on anthropometry, body composition and adherence to the Mediterranean diet in women with systemic lupus erythematosus.

	Change from Baseline at Week 12 (Final–Baseline)							
	Exercise (*n* = 26)		Comparison (*n* = 32)		Exercise vs. Comparison			
	Mean	SE	Mean	SE	Mean Difference	(95% CI)		*p*
Weight (kg)	−0.5	0.4	−1.3	0.4	0.9	−0.3	2.0	0.126
BMI (kg/m^2^)	−0.3	0.2	−0.5	0.1	0.2	−0.2	0.7	0.279
Waist circumference (cm)	1.0	1.3	−3.1	1.2	4.1	0.5	7.7	0.026 *
Waist-to-hip ratio	0.0	0.0	0.0	0.0	0.0	0.0	0.0	0.927
Fat percentage	−1.4	0.7	−1.6	0.7	0.2	−1.8	2.2	0.849
Lean mass (kg)	−0.1	0.3	0.2	0.3	−0.3	−1.1	0.6	0.535
Lean mass percentage	0.2	0.5	1.0	0.5	−0.8	−2.3	0.6	0.245
Adherence to the Mediterranean Diet (0–55)	1.6	1.0	−0.6	0.9	2.1	−0.7	4.9	0.131

The analyses were adjusted for baseline values. BMI, Body Mass Index; CI, confidence interval; SE, standard error. * *p* < 0.05.

**Table 4 nutrients-15-04424-t004:** Sensitivity analyses: effects of a 12-week aerobic training intervention on anthropometry, body composition and adherence to the Mediterranean diet in women with systemic lupus erythematosus (participants in the exercise group were included if attendance was ≥90%).

	Change from Baseline at Week 12 (Final–Baseline)							
	Exercise (*n* = 18)		Comparison (*n* = 28)		Exercise vs. Comparison			
	Mean	SE	Mean	SE	Mean Difference	(95% CI)		*p*
Weight (kg)	−1.0	0.5	−1.5	0.4	0.5	−0.9	1.8	0.493
BMI (kg/m^2^)	−0.4	0.2	−0.5	0.2	0.1	−0.4	0.6	0.720
Waist circumference (cm)	−0.4	1.6	−3.5	1.3	3.0	−1.2	7.3	0.156
Waist-to-hip ratio	0.0	0.0	0.0	0.0	0.0	−0.1	0.0	0.416
Fat percentage	−1.5	0.9	−1.8	0.7	0.3	−2.1	2.6	0.810
Lean mass (kg)	−0.1	0.4	0.2	0.4	−0.3	−1.5	0.8	0.557
Lean mass percentage	0.5	0.7	1.1	0.6	−0.6	−2.4	1.2	0.506
Adherence to the Mediterranean Diet (0–55)	1.5	1.4	−0.3	1.1	1.9	−1.8	5.5	0.307

The analyses were adjusted for baseline values. BMI, Body Mass Index; CI, confidence interval; SE, standard error.

## Data Availability

Data available on request from the authors.

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
