# Peer review of "No Changes in Body Composition and Adherence to the Mediterranean Diet after a 12-Week Aerobic Training Intervention in Women with Systemic Lupus Erythematosus: The EJERCITA-LES Study"

_nutrients, 2023, doi:10.3390/nu15204424_

Round 1

Reviewer 1 Report

The authors conducted a secondary analysis to examine the effect of a 12-week aerobic exercise training on body composition parameters in women with systemic lupus erythematosus. They found that the 12-week exercise training did not induce any significant changes in body composition parameters in the patients.  Overall, the manuscript is well written and easy to follow. At the same time, the manuscript has several outstanding issues to be further addressed.

Majors)

This study has several critical flaws.

First, the authors need to provide research hypothesis toward the end of the Introduction along with their logic to support the hypothesis.

Second, this study does not appear to control and/or monitor potential exposures (such as daily physical activity and caloric intake) carefully enough. Since weight change reflects the balance of energy intake and expenditure, monitoring daily dietary intakes and physical activities and sedentary behaviors such as sitting time (energy expenditure) would provide critical information in interpreting the outcomes.

Third, I am afraid that the conclusion is not supported by the data of the study. It appears that the authors failed to test the research hypothesis (although the research hypothesis is not stated in the Introduction).

Minors)

-      In Table 3, the authors need to provide a clear explanation for no change difference in WHR even with a significant change difference in waist circumference: why did the control group but not the exercise group have a significant reduction in waist circumference?

-      As mentioned in Discussion, the primary aim of the study was not to examine the effect of exercise intervention on body composition. Therefore, power analysis for sample size calculation should be done using the body composition changes as major outcomes. If this is the case, the authors failed to test their research hypothesis.

-      Conducting the sensitivity analysis (intention-to-treat analysis) is enough (I think that table 4 is not necessary).

Author Response

ANSWER TO REVIEWERS’ COMMENTS

Manuscript Number: nutrients-2649047.

Dear Editor,

Please, find enclosed the revised version of our manuscript entitled “No changes in body composition and adherence to the Mediterranean diet after a 12-week aerobic training intervention in women with systemic lupus erythematosus: the EJERCITA-LES study” by Gavilán-Carrera et al. to be considered for publication in Nutrients. We would like to thank the Editor and the Reviewers for their thoughtful and constructive comments and for giving us the opportunity to improve the quality of our manuscript. We have carefully considered all of the suggestions, and have either integrated them into the revised manuscript or provided our rationale for not doing so. Changes to the original manuscript are highlighted in yellow. We believe that our manuscript has been improved as a result of these modifications. An itemized point-by-point response to the Reviewer’ comments is presented below.

Reviewer 1

Comment: The authors conducted a secondary analysis to examine the effect of a 12-week aerobic exercise training on body composition parameters in women with systemic lupus erythematosus. They found that the 12-week exercise training did not induce any significant changes in body composition parameters in the patients.  Overall, the manuscript is well written and easy to follow. At the same time, the manuscript has several outstanding issues to be further addressed.

Response:  Thank you very much for reviewing our work and for your constructive feedback and evaluation.

Comment: First, the authors need to provide research hypothesis toward the end of the Introduction along with their logic to support the hypothesis.

Response: The hypotheses of the study have been included at the end of the introduction (lines 89-91). Thank you very much for you comment.

Comment: Second, this study does not appear to control and/or monitor potential exposures (such as daily physical activity and caloric intake) carefully enough. Since weight change reflects the balance of energy intake and expenditure, monitoring daily dietary intakes and physical activities and sedentary behaviors such as sitting time (energy expenditure) would provide critical information in interpreting the outcomes.

Response:  We completely agree with the reviewer on the relevance of accounting for energy intake and expenditure. Unfortunately, this study was not originally designed to measure these variables, as its primary focus was not to investigate changes in body composition parameters. This limitation has been highlighted in the discussion (lines 330-336 and 363-365); we have additionally emphasized that addressing this limitation presents an opportunity for future research (lines 377-379).

Comment: Third, I am afraid that the conclusion is not supported by the data of the study. It appears that the authors failed to test the research hypothesis (although the research hypothesis is not stated in the Introduction).

Response: The conclusions have been revised and modified according to the hypothesis proposed (abstract, lines 282-284, and 392)

Comment: In Table 3, the authors need to provide a clear explanation for no change difference in WHR even with a significant change difference in waist circumference: why did the control group but not the exercise group have a significant reduction in waist circumference?

Response: Thank you very much for raising this relevant point. According to previous evidence [1], it is difficult to distinguish clinically relevant changes from measurement bias when it comes to waist circumferences due to large measurement errors. While attempts were made to standardize data collection within this study, it should be noted that extensive training is required to reduce measurement errors [1]. Therefore, the possibility of errors and discrepancies among evaluators cannot be completely ruled out. This measurement error might be of a lesser extent in hip circumference measures [2], which could explain the inconsistency in findings related to waist circumference when compared to waist to hip ratio. This information has been included in lines 301-307.

References

  1. Verweij, L.M.; Terwee, C.B.; Proper, K.I.; Hulshof, C.T.; Mechelen, W. Van Measurement error of waist circumference: Gaps in knowledge. Public Health Nutr. 2013, 16, 281–288.
  2. Sebo, P.; Herrmann, F.R.; Haller, D.M. Accuracy of anthropometric measurements by general practitioners in overweight and obese patients. BMC Obes. 2017, 4, 1–7.

Comment: As mentioned in Discussion, the primary aim of the study was not to examine the effect of exercise intervention on body composition. Therefore, power analysis for sample size calculation should be done using the body composition changes as major outcomes. If this is the case, the authors failed to test their research hypothesis.

Response: The power achieved in the analyses included in the present study was calculated for each outcome of body composition and adherence to the Mediterranean diet. This sentence has been rewritten in lines 204-207 to improve clarity.

Comment: Conducting the sensitivity analysis (intention-to-treat analysis) is enough (I think that table 4 is not necessary).

Response: Thank you very much for this suggestion. We aimed to analyze the consistency of the findings across various sensitivity analyses, with each of them providing distinct and relevant information: intention-to-treat analysis provided information related to effectiveness whereas per-protocol informed about efficacy of the program. The authors consider that the results displayed in table 4 (per-protocol analysis requiring attendance>90%) help to better understand to what extent the level of attendance influence the efficacy of the program. In order to facilitate interpretation of the results, we decided to maintain all the sensitivity analyses in the main manuscript. Nevertheless, we are open to moving this content to supplementary material if it is considered to be more appropriate.

Reviewer 2 Report

It was a great pleasure that I reviewed the manuscript entitled “No changes in body composition and adherence to the Mediterranean diet after a 12-week aerobic training intervention in women with systemic lupus erythematosus: the EJERCITA-LES study”. I think the paper presents some interesting findings. However, the paper requires several methodological clarifications. Here are several comments and suggestions that might help strengthen the paper.

1.     I think the Introduction section is well-written, but the authors should provide some explanations why the aerobic training intervention could also influence adherence to the Mediterranean diet.  

2.     In the Methods section, I would suggest the use of the “Comparison Group” instead of “Control Group” as the study did not employ random assignment. Also, please clarify what the authors meant by “basic nutritional recommendations (lines 116-117).”

3.     As in participants in the “Control Group”, I believe participants in the Experimental Group were also given basic nutritional recommendations  for the Mediterranean diet. Please clarify.

4.     In Lines 173 to 177, the authors provided some justifications why the random assignment were not suable for this study. I think it is important to provide more details – for instance, this means that participants in the Exercise condition are only those lived close to the hospital?

5.     I was wondering if lines 217 to 223 and Figure 2 are really necessary.

6.     What I was most impressed reported in the Results section is that adherence to the Mediterranean diet was actually slightly increased across all analyses, although the effects did not reach conventional levels of significance. I think this is something worth noting in the Discussion section instead of just addressing this as non-significat effect compared to the previous studies.

7.     More generally, I think it might be very difficult to observe the effects of any exercise interventions on weight or BMI in just 3 months or so. Along with the above finding, another point I think worth noting as a future direction is examining the effects of even longer exercise interventions.

Author Response

ANSWER TO REVIEWERS’ COMMENTS

Manuscript Number: nutrients-2649047.

Dear Editor,

Please, find enclosed the revised version of our manuscript entitled “No changes in body composition and adherence to the Mediterranean diet after a 12-week aerobic training intervention in women with systemic lupus erythematosus: the EJERCITA-LES study” by Gavilán-Carrera et al. to be considered for publication in Nutrients. We would like to thank the Editor and the Reviewers for their thoughtful and constructive comments and for giving us the opportunity to improve the quality of our manuscript. We have carefully considered all of the suggestions, and have either integrated them into the revised manuscript or provided our rationale for not doing so. Changes to the original manuscript are highlighted in yellow. We believe that our manuscript has been improved as a result of these modifications. An itemized point-by-point response to the Reviewer’ comments is presented below.

Reviewer 2

Comment: It was a great pleasure that I reviewed the manuscript entitled “No changes in body composition and adherence to the Mediterranean diet after a 12-week aerobic training intervention in women with systemic lupus erythematosus: the EJERCITA-LES study”. I think the paper presents some interesting findings. However, the paper requires several methodological clarifications. Here are several comments and suggestions that might help strengthen the paper.

Response: Thank you very much for reviewing our work and for the suggestions to improve it.

Comment: I think the Introduction section is well-written, but the authors should provide some explanations why the aerobic training intervention could also influence adherence to the Mediterranean diet. 

Response: Additional information regarding how physical activity interventions can lead to modifications in dietary preferences close to the Mediterranean diet has been added to the introduction (lines 79-86). Thank you.

Comment: In the Methods section, I would suggest the use of the “Comparison Group” instead of “Control Group” as the study did not employ random assignment. Also, please clarify what the authors meant by “basic nutritional recommendations (lines 116-117).”

Response: Thank you for this suggestion. The term “comparison group” is now used in the manuscript. We have specified in lines 120-125 that: “…the patients assigned to the Comparison Group (usual care) were given basic guidelines to adopt a healthy lifestyle as normally delivered in standard treatment. This included basic nutritional recommendations (i.e. reducing processed foods and increasing intake of vegetables / fruits) and physical activity advice (i.e. reducing sedentary behavior and increasing daily levels of physical activity)”.

Comment: As in participants in the “Control Group”, I believe participants in the Experimental Group were also given basic nutritional recommendations for the Mediterranean diet. Please clarify.

Response: We have clarified in lines 127-129 that: “Patients assigned to the exercise group received identical guidelines for adopting a healthy lifestyle (i.e. basic nutritional recommendations and advice on physical activity), just like those in the comparison group.

Comment: In Lines 173 to 177, the authors provided some justifications why the random assignment were not suable for this study. I think it is important to provide more details – for instance, this means that participants in the Exercise condition are only those lived close to the hospital?

Response: Thank you for raising this point. Accordingly, we have extensively explained this point in lines 184-188: “Different factors, primarily the distance from the hospital, as well as other personal reasons related to time availability (work and/or domestic responsibilities), prevented a significant number of participants from regularly attending the intervention over the 12-week period. Therefore, in order to maximize sample size, randomization was not finally possible. To minimize selection bias, the groups were matched by age (± 2 years), BMI (± 1 kg/m²), and SLEDAI (± 1 unit).” It is also worth noting that the trial quality may have a greater impact on the treatment effect size than the randomization alone, and non-randomized controlled studies of high quality can produce outcomes that approximate those found in randomized controlled trials [3]. This information has been also included in the limitation and strength section (lines 365-369).

References

  1. Ferriter, M.; Huband, N.; Healthcare, N. Does the non-randomized controlled study have a place in the systematic review ? A pilot study. 2005, 111–120.

Comment: I was wondering if lines 217 to 223 and Figure 2 are really necessary.

Response: We completely understand the reviewer’s concern as the results shown in figure 2 are secondary. However, we believe they provide relevant and interesting information to better understand whether the exercise intervention initially planned was finally met in terms of intensity, which can greatly influence the final effects on body composition parameters. Reporting these training parameters is essential for a proper interpretation of the effects of various exercise doses, and they are often absent in the exercise literature [4]. Then, the authors decided to maintain the presentation of these findings, although they have been moved to supplementary material to improve readability of the manuscript.

References

  1. Slade, S.C.; Dionne, C.E.; Underwood, M.; Buchbinder, R. Consensus on Exercise Reporting Template (CERT): explanation and elaboration statement. Br J Sport. Med 2016, 50, 1428–1437.

Comment: What I was most impressed reported in the Results section is that adherence to the Mediterranean diet was actually slightly increased across all analyses, although the effects did not reach conventional levels of significance. I think this is something worth noting in the Discussion section instead of just addressing this as non-significant effect compared to the previous studies.

Response:  The interpretation of these findings has been extended in lines 346-354. Thank you for raising this point. “… It is worth mentioning that in the present study adherence to the Mediterranean diet was slightly and consistently enhanced in the exercise group, although this increase did not reach statistical significance. Given that previous research has shown that physical activity interventions can indirectly increase fruit and vegetable consumption [23–26], it is possible that exercise primarily influences specific food groups within the Mediterranean Diet, rather than the diet as a whole. Therefore, it could be hypothesized that: i) our exercise intervention could be a stimulus that only produces slight but insufficient modifications in dietary patterns and/or ii) Mediterranean Diet could be a very specific pattern to be indirectly modified through exercise.”

Comment: More generally, I think it might be very difficult to observe the effects of any exercise interventions on weight or BMI in just 3 months or so. Along with the above finding, another point I think worth noting as a future direction is examining the effects of even longer exercise interventions.

Response: Thank you very much for this suggestion. The need for future intervention studies analyzing longer exercise and nutritional interventions has been included in lines 382-385.

Round 2

Reviewer 1 Report

I am satisfied with the author's reply to my comments.

Congratulations!